# Natural Killer Cell Activation Receptor NKp30 Oligomerization Depends on Its *N*-Glycosylation

**DOI:** 10.3390/cancers12071998

**Published:** 2020-07-21

**Authors:** Ondřej Skořepa, Samuel Pazicky, Barbora Kalousková, Jan Bláha, Celeste Abreu, Tomáš Ječmen, Michal Rosůlek, Alexander Fish, Arthur Sedivy, Karl Harlos, Jan Dohnálek, Tereza Skálová, Ondřej Vaněk

**Affiliations:** 1Department of Biochemistry, Faculty of Science, Charles University, Hlavova 2030, 12840 Prague, Czech Republic; ondrej.skorepa@natur.cuni.cz (O.S.); spazicky@gmail.com (S.P.); barbora.kalouskova@natur.cuni.cz (B.K.); jahabla@gmail.com (J.B.); celesteabreu22@gmail.com (C.A.); tomas.jecmen@natur.cuni.cz (T.J.); rosulek.michal@gmail.com (M.R.); 2BIOCEV, Institute of Microbiology, The Czech Academy of Sciences, Průmyslová 595, 25250 Vestec, Czech Republic; 3Department of Biochemistry, Oncode Institute, Netherlands Cancer Institute, Plesmanlaan 121, 1066 CX Amsterdam, The Netherlands; a.fish@nki.nl; 4Protein Technologies, Vienna Biocenter Core Facilities GmbH, Dr. Bohr-Gasse 3, 1030 Vienna, Austria; arthur.sedivy@vbcf.ac.at; 5Division of Structural Biology, Wellcome Centre for Human Genetics, University of Oxford, Roosevelt Drive, Oxford OX3 7BN, UK; karl@strubi.ox.ac.uk; 6BIOCEV, Institute of Biotechnology, The Czech Academy of Sciences, Průmyslová 595, 25250 Vestec, Czech Republic; dohnalek@ibt.cas.cz (J.D.); t.skalova@gmail.com (T.S.)

**Keywords:** NK cell, NKp30, B7-H6, glycosylation, oligomerization

## Abstract

NKp30 is one of the main human natural killer (NK) cell activating receptors used in directed immunotherapy. The oligomerization of the NKp30 ligand binding domain depends on the length of the C-terminal stalk region, but our structural knowledge of NKp30 oligomerization and its role in signal transduction remains limited. Moreover, ligand binding of NKp30 is affected by the presence and type of *N*-glycosylation. In this study, we assessed whether NKp30 oligomerization depends on its *N*-glycosylation. Our results show that NKp30 forms oligomers when expressed in HEK293S GnTI^−^ cell lines with simple *N*-glycans. However, NKp30 was detected only as monomers after enzymatic deglycosylation. Furthermore, we characterized the interaction between NKp30 and its best-studied cognate ligand, B7-H6, with respect to glycosylation and oligomerization, and we solved the crystal structure of this complex with glycosylated NKp30, revealing a new glycosylation-induced mode of NKp30 dimerization. Overall, this study provides new insights into the structural basis of NKp30 oligomerization and explains how the stalk region and glycosylation of NKp30 affect its ligand affinity. This furthers our understanding of the molecular mechanisms involved in NK cell activation, which is crucial for the successful design of novel NK cell-based targeted immunotherapeutics.

## 1. Introduction

Natural killer (NK) cells display spontaneous cytotoxic activity without prior sensitization in a process known as natural cytotoxicity. Accordingly, this subset of lymphocytes plays a key role in early immune defense within innate immunity [1,2]. In particular, mature NK cells recognize tumor cells and certain virus-infected cells through a number of inhibitory and activating receptors [3]. The inhibitory receptors bind to human leukocyte antigen (HLA) molecules, which block NK cell cytotoxicity, whereas their activating receptors bind to specific molecules expressed on target cells, which activate NK cells. Hence, NK cell activation depends on the balance between inhibitory and activating signals coming from their inhibitory and activating receptors [3,4], which are classified primarily into two families: C-type lectin-like and immunoglobulin-like receptors [5].

Natural cytotoxicity receptors (NCRs), namely the natural killer cell proteins NKp30, NKp44, and NKp46 (numbers refer to their molecular weights), stand out for their role in activating NK cells and initiating tumor targeting [5]. NCRs are type I transmembrane proteins that consist of one or two extracellular immunoglobulin-like (Ig-like) domains, a transmembrane α-helix with a positively charged amino acid that facilitates the interaction with signaling adaptor proteins, and a short C-terminal intracellular chain [4].

Natural killer protein 30 (NKp30; also known as natural cytotoxicity receptor 3, NCR3; or CD337) is an Ig-like activating receptor of NK cells [4,6]. Similar to other members of CD28 protein family, the extracellular part of NKp30 consists of a single N-terminal Ig-like domain, followed by a distinct 15 amino acids long stalk region proximal to the plasma membrane, which is important for receptor signaling [7]. Most CD28 family members bind proteins of the B7 family [8]; however, the first molecules that have been shown to interact with NKp30 are heparin and heparin sulfate, that serve as co ligands and only interact with the glycosylated receptor [4,9]. On the other hand, interaction of NKp30 with viral ligands, such as hemagglutinin of the ectromelia and vaccinia virus [10] and protein pp65 released from human cytomegalovirus [10], has inhibitory effects on NK cells [4]. For example, pp65 binding to NKp30 causes NKp30-CD3ζ complex dissociation, thus disrupting the activating signaling pathway, albeit without preventing other ligands from binding to NKp30 because pp65 uses another binding site [4,11].

Further, three specific cellular ligands of NKp30 have been identified. Consistent with the other members of CD28 family, NKp30 binds a member of B7 family, in particular B7 homolog 6 (B7-H6), that is constitutively expressed on the surface of some tumor cells [12]. Another NKp30 cellular ligand, BAG-6, plays a role in DNA damage response, gene expression regulation, protein quality control, and immunoregulation in healthy cells [13], but is recruited to the cell membrane and interacts with NKp30 in some tumor cells or under the stress conditions [13]. Both B7-H6 and BAG-6 also exist on the surface of exosomes [13] and in soluble forms that result from proteolytic shedding of their membrane forms, which hinders NK cell activation [13,14]. Lastly, the most recently discovered NKp30 ligand, galectin 3, expressed on the surface of some tumors, almost completely blocks NK cell cytotoxicity [15].

Since the B7-H6:NKp30 interaction was first discovered, considerable research efforts have been made to define B7-H6 expression specificity in tumor cell lines and its underlying clinical potential [8,16,17]. Similarly to NKp30, B7-H6 is a type I transmembrane protein [12]. It consists of two extracellular Ig-like domains (IgV and IgC), an α-helical transmembrane domain, and a C-terminal sequence homologous to group-specific antigen (GAG) proteins. This C-terminal sequence has various signaling motifs, including ITIM-, SH-2-, and SH-3-binding motifs [8,12], which trigger signal transduction upon NKp30 binding. However, the signal outcome remains unknown [8]. The closest structural homologs of B7-H6 are B7-H1 (more widely known as programmed death-ligand 1 or PD-L1) and B7-H3, all of which belong to the B7 protein family [8,12].

B7-H6 is expressed in several tumor cell lines, both ex vivo and in vivo, but remains undetected in healthy cells thus far, and is therefore an excellent tumor marker [8,12]. In addition, B7-H6 expression has been induced in monocytes and neutrophils [18] using agonists of Toll-like receptors (TLR2, TLR4, TLR5, and TLR8), interleukin IL-1β, and tumor necrosis factor alpha (TNFα), with consistent B7-H6 mRNA and protein expression kinetics. In these phagocytes, B7-H6 mRNA expression peaked early, from 3 to 12 h after induction, returning to baseline within 24 h, whereas its surface expression was stable for up to 48 h after induction. Soluble or exosomal B7-H6 expression has also been induced using the same factors. In in vitro studies, both soluble and membrane forms of B7-H6 were detected in the blood of patients with systemic inflammatory response syndrome, but its cell-surface expression was only identified in patients presenting with sepsis and selective for CD14^+^ and CD161^+^ pro-inflammatory monocytes. In contrast, the blood of patients presenting with sepsis caused by Gram-negative bacteria contains soluble or exosomal forms of B7-H6 [18].

Although crystal structures of NKp30, both unbound [19] and in a complex with B7-H6 [20], have been solved, these structures refer to NKp30 recombinantly expressed in bacteria and therefore lacking glycosylation. However, two *N*-glycosylation sites of NKp30 are crucial for its signal transduction and for B7-H6 but not BAG-6 ligand binding [21]. Moreover, NKp30 lacks the C-terminal 15 amino acids long stalk region that connects the ligand binding domain to the transmembrane helix in these structures. Importantly, both glycosylation and the stalk region affect the binding affinity of NKp30 [22]. The stalk region affects the oligomeric state of NKp30; NKp30 without the stalk region forms only monomer and dimer species in solution, whereas NKp30 with the stalk region forms higher oligomers [21]. Interestingly, the NKp30 dimer is observed in the crystal structure of its unbound (construct without the stalk region produced in *Escherichia coli*) [19] but not in that of its B7-H6-bound state [20]. Nevertheless, the presence of oligomers is positively correlated with NKp30 affinity to its ligands, as previously measured by surface plasmon resonance (SPR), although the differences in the measured affinity are rather small. This increased affinity is due to an avidity effect, which is attributed to NKp30 ectodomain oligomerization [21]. Greater changes in affinity are caused by alterations in *N*-glycosylation; glycosylation at Asn42 is essential for ligand binding, and glycosylation at Asn68 also has a substantial effect, whereas glycosylation at Asn121 does not play a key role, as determined by site-directed mutagenesis [22].

Furthermore, the analysis of NKp30 ligand binding has highlighted differences in affinity as a function of the protein expression system used. NKp30 affinity to B7-H6 ranged from 2.5 to 3.5 μM with both recombinant proteins expressed in *Escherichia coli* [19]. However, this affinity was higher (1 μM) when B7-H6 was expressed in the Sf9 insect cell line [20] and even higher (ranging from 80 to 320 nM) when NKp30-Ig fusion protein was expressed in the human cell line HEK293T (which provides a complex *N*-glycosylation pattern). In the latter case, the affinity of NKp30 to B7-H6 additionally increased with longer stalk region [22]. Even higher affinities were recorded when expressing NKp30 in Sf9 cells and B7-H6 in HEK293T cells, assessing affinities ranging from 1 to 2 nM by ELISA, depending on the oligomeric state of NKp30 [21]. Interestingly, the affinity of NKp30 expressed in HEK293T cells to BAG-6 isolated from an insect cell line was 64 nM. In contrast, its affinity to BAG-6 isolated from *Escherichia coli* was two times higher [23].

In this study, we show that NKp30 oligomerization depends on its *N*-glycosylation. NKp30 produced in HEK293S GnTI^−^ cells lacking *N*-acetylglucosaminyl-transferase I activity forms oligomers but disassembles to pure monomers after enzymatic deglycosylation. We have further characterized the binding affinity of B7-H6 to NKp30 and its dependence on glycosylation and oligomeric state using proteins expressed in human cell lines, that closely mimic the natural post-translational modifications of this receptor. Finally, we have determined the crystal structure of the NKp30:B7-H6 complex with the glycosylated receptor. Our results suggest that dimerization may be a necessary step for NKp30 oligomerization and stable signal transduction upon B7-H6 binding.

## 2. Results

### 2.1. Recombinant Expression and Purification of Stable, Soluble, Glycosylated NKp30, and B7-H6 Proteins

For our studies, we cloned the extracellular domains of NKp30 and B7-H6 into a mammalian expression vector with a C-terminal histidine tag. Both proteins were expressed either in HEK293T cell line that provides the protein with complex wild-type *N*-glycans, or in HEK293S GnTI^−^ cell line that provides uniform, simple Asn-GlcNAc_2_Man_5_
*N*-glycans, in the latter case allowing also for a possibility of efficient enzymatic deglycosylation, when required [24,25,26]. Presence of the expected glycan type was verified by mass spectrometry (Appendix A). Two constructs were prepared to study the effect of the C-terminal stalk region of the NKp30 extracellular domain. The NKp30_Stalk construct contains the whole NKp30 extracellular domain, including the stalk region, whereas the NKp30_LBD (Ligand Binding Domain) lacks the stalk sequence, which is the C-terminal section of the extracellular domain of NKp30 (Figure 1). The entire extracellular portion of B7-H6 (Asp25–Leu245) consisting of two Ig-like domains was used. Each Ig-like domain contains one disulfide bridge, and moreover, the C-terminal domain contains one odd cysteine residue (Cys212).

B7-H6 expression in HEK293 cells yielded 5 mg of purified protein per liter of cell culture. However, when we analyzed fractions resulting from the SEC peak (Figure 2a, red line) by SDS-PAGE, we noticed that the protein appeared as a monomer of expected size in reducing buffer (33 kDa + *N*-glycosylation), but in non-reducing buffer, a band with a size corresponding to the B7-H6 dimer was identified (Figure 2b, left side, asterisk). These results suggest that the protein formed covalent dimers via its odd cysteine residue. Sedimentation analysis also showed that, although most of the protein behaved as monomers, sedimenting at 2.5 S, a small amount of putative dimer (4 S) and tetramer (6.1 S) species were also present (Figure 2c). The average fitted *f*/*f*_0_ frictional ratio of 1.5 indicates an elongated shape of the molecule, in line with the published structure. However, mass spectrometry analysis confirmed that disulfide bridges of this wild-type B7-H6 expression construct were not linked correctly (Appendix A).

The odd cysteine that forms the unwanted covalent dimer of B7-H6 was identified as C212 according to the published structure (PDB 3PV6) [20]; therefore, we mutated this cysteine to serine (C212S). This mutation resulted in the proper folding of B7-H6 (Appendix A), and strikingly, promoted the expression yield up to 50 mg per liter of cell culture (Figure 2a, black line). In the present study, only this C212S mutated form of B7-H6 was used and is henceforth referred to as B7-H6 for simplicity. The expression of both NKp30 constructs in HEK293 cell lines was straightforward, yielding 40 and 23 mg of NKp30_LBD, and 27 and 14 mg of NKp30_Stalk per liter of cell culture when expressed in HEK293T and HEK293S GnTI^−^ cell line, respectively.

### 2.2. Protein Deglycosylation

To assess the effect of glycosylation on NKp30 oligomerization and on its binding properties, we deglycosylated both NKp30_Stalk and NKp30_LBD. Moreover, as B7-H6 has six predicted *N*-glycosylation sites (Figure 1), the complexity of wild-type mammalian glycosylation might hinder crystallization and therefore we deglycosylated also B7-H6. Additionally, we also assessed the effect of the type of B7-H6 glycosylation on NKp30 receptor binding. The deglycosylated proteins were analyzed by SDS-PAGE (Figure 3).

Under standard conditions (10 mM HEPES pH 7.5, 150 mM NaCl, 10 mM NaN_3_), B7-H6 deglycosylation caused its precipitation, most likely because of the loss of all its glycans. For this reason, we sought to improve the buffer conditions for deglycosylation and the storage of this protein by differential scanning fluorimetry. Twenty-five conditions were screened by analyzing protein melting point temperatures (Figure 4). The initial melting temperature *T_m_* of the protein in the HEPES buffer was 50 °C and changing the pH of the buffer had no effect. Similarly, adding salts or stabilizers such as L-Arginine had virtually no effect either. Conversely, the highest *T_m_* was recorded when adding 0.5 M saccharose (58 °C). However, at 20% glycerol, the *T_m_* increased similarly (56 °C) and B7-H6 aggregation was completely prevented during deglycosylation. Therefore, glycerol was added to a final concentration of 20% (*v*/*v*) for convenience and used for further B7-H6 deglycosylation and storage at low temperatures (−20 °C). Glycerol was always removed by buffer exchange on desalting columns prior to subsequent experiments.

### 2.3. NKp30 Glycosylation Promotes Its Oligomerization

The extracellular part of the NKp30 receptor has three *N*-glycosylation sites (Figure 1). In previous study, the presence of its *N*-glycans has been shown to enhance B7-H6 ligand binding [22] and associated cell signaling. In addition, NKp30 also forms non-covalent oligomers when the construct contains the C-terminal stalk region [21]. Accordingly, we determined the size of these oligomers by analytical ultracentrifugation (AUC) and SEC with multi-angle light scattering (MALS) detection. Notably, the NKp30_LBD construct lacking the stalk region is purely monomeric, whereas the NKp30_Stalk construct is present in both monomeric and oligomeric species (Figure 5a). Surprisingly, we could show that not only the stalk region, but also *N*-glycosylation is essential for the formation of NKp30 oligomers. Whereas NKp30_Stalk expressed in HEK293T or HEK293S GnTI^−^ cell line contains a prominent fraction of oligomers, the deglycosylation of NKp30 with Endo F1 (leaving a single GlcNAc unit at the glycosylation site) completely depletes the sample of oligomers (Figure 5b). Elution profile of concentrated oligomeric fraction of NKp30_Stalk shows that most of the protein remains in the oligomeric form and only a minor fraction dissociates to monomers, highlighting slow kinetics of the oligomer dissociation. However, deglycosylation of the same oligomeric fraction again depletes the oligomers and only monomeric protein remains in the sample.

We repeated the same experiment and analyzed it using analytical ultracentrifugation, which led to the same conclusion; NKp30_Stalk does not form oligomers when deglycosylated (Figure 5c). Sedimentation analysis provided better resolution of the oligomeric species than SEC and thus allowed us to estimate their size. The main peak at 1.8 S, with a predicted molar mass of 18.8 kDa, matches the expected mass of the glycosylated NKp30_Stalk monomer (18.9 kDa). The molar masses predicted for the peaks corresponding to the oligomeric species indicated the presence of oligomers with 3, 5, 10, and 20 units, on average. The fitted *f*/*f*_0_ ratio of 1.6 suggests that the oligomers have a considerably elongated or flattened shape. Interestingly, NKp30_Stalk with wild-type glycosylation expressed in HEK293T cells (20–25 kDa) showed a more homogeneous oligomer profile with one dominant species of ca 8 units and with secondary species of ca 16 units, based on their calculated molar masses of 180 and 320 kDa, respectively. These findings corroborate the molar masses calculated from the MALS signal (Figure 5b), which are within the same range. In summary, not only the presence of the stalk domain but also the glycosylation is essential for the formation of NKp30 oligomers.

### 2.4. B7-H6 Forms Equimolar Complex with Monomeric NKp30, But Not with Its Oligomeric Form

To study the impact of B7-H6 on NKp30 oligomers, we followed the NKp30:B7-H6 interaction using hydrodynamic approaches. NKp30_Stalk and B7-H6 were analyzed by SEC-MALS separately and in a complex at a 1:1 molar ratio (Figure 6a). Shifts in the elution volume and in the calculated molar mass suggest the formation of the complex in the monomeric and oligomeric fractions of the NKp30_Stalk construct. The ratio between the areas of monomeric and oligomeric peaks did not change, indicating that B7-H6 binding does not disrupt or induce NKp30 oligomerization. Subsequently, we analyzed the binding of the monomeric and oligomeric fractions separately by AUC. The monomeric fraction of glycosylated NKp30_Stalk sedimented predominantly at 1.7 S (monomer) and marginally at 3.5 S (dimeric or trimeric species). B7-H6 sedimented as monomer at 2.4 S. The equimolar mixture of these proteins sedimented as a 1:1 complex at 3.1 S, with a relatively small peak observed at 1.7 S, which is most likely an excess of the free monomer of NKp30 (Figure 6b).

To monitor the binding of the oligomeric fraction, we labelled B7-H6 with NHS-ATTO488 fluorescent dye and performed the experiment at 280 nm (total signal) and 480 nm (B7-H6 only) wavelengths (Figure 6c). The bimodal distribution of NKp30_Stalk HEK293T oligomers changed to a single oligomeric complex upon the addition of B7-H6, with a sedimentation coefficient of 11.2 S that corresponds to ca 340 kDa at the fitted overall *f*/*f*_0_ ratio of 1.7. This value is lower than expected for the fully saturated NKp30_Stalk_HEK293T putative octamer (ca 180 kDa plus eight times 30–33 kDa for each glycosylated B7-H6, i.e., approximately 420–440 kDa for the fully saturated 1:1 complex). This suggests that not all NKp30 binding sites in the oligomeric species are accessible for B7-H6 binding and that NKp30 molecules might sterically block the access to the neighboring binding sites within the oligomer itself or with the B7-H6 molecules bound to it. The modest increase in MALS-calculated molar mass for the oligomeric NKp30:B7-H6 complex (Figure 6a) supports these inferences. Indeed, the ratio between the areas of monomeric and oligomeric NKp30:B7-H6 differs between 280 nm and 480 nm, which indicates that the amount of B7-H6 present in oligomers is smaller than the amount of NKp30 in oligomers (Figure 6c). Finally, the areas under the oligomeric peak are not equal between the two wavelengths, suggesting that only approximately 60% of the NKp30 binding sites are occupied by B7-H6 in the oligomers. This matches well with the observed mass difference between the oligomeric complex (340 kDa) and the oligomers (180 kDa), resulting in ca 160 kDa of B7-H6 bound to the oligomers, or approximately five B7-H6 molecules per eight NKp30 molecules in the oligomeric complex. Hence, B7-H6 binds both monomers and oligomers of NKp30, but only sub-equimolar amount of NKp30 binding sites is available for B7-H6 binding in the oligomeric fraction.

### 2.5. Affinity of NKp30:B7-H6 Interaction Differs between Surface and Solution

To reevaluate the impact of glycosylation on the interaction of B7-H6 and NKp30, we measured the affinity of this interaction using proteins with different glycosylation patterns. B7-H6 has six predicted *N*-glycosylation sites, five of which were confirmed by our MS analysis (Appendix A). B7-H6 in wild-type (T), simple (S), and Endo F1-deglycosylated (D) glycosylation states were immobilized on the SPR chip. The same analyte, the monomeric fraction of NKp30_Stalk, was measured in individual cells. Neither the dissociation constant *K_D_* nor the maximal response *B_max_* values significantly differed among all tested B7-H6 variants (Figure 7a). Therefore, B7-H6 glycosylation does not significantly affect the NKp30 receptor binding.

Subsequently, we investigated the effect of NKp30 deglycosylation on B7-H6 ligand binding. A previous study had already shown that the point mutation of the glycosylation site Asn68 (N68Q) reduces the affinity for the B7-H6 ligand, whereas the N42Q mutation of the Asn42 glycosylation site almost completely abrogates binding [22]. For this reason, we performed a binding assay with NKp30 deglycosylated with Endo F1. Interestingly, in comparison with the aforementioned study, the affinity of both deglycosylated NKp30_LBD and NKp30_Stalk did not significantly differ from that of their wild-type or uniformly glycosylated counterparts in both SPR and isothermal titration calorimetry (ITC) measurements (Figure 7b–e). In fact, for NKp30_LBD, the affinity even moderately increased from uniform glycans to wild-type glycans to deglycosylated protein (Figure 7b,d). Similarly, only slight differences in affinity were found among NKp30_Stalk glycosylation variants (Figure 7c,e). However, Endo F1 treatment leaves a single GlcNAc unit at each glycosylation site, whereas point mutations completely block glycosylation. Hence, these results cannot be compared directly.

In contrast, NKp30_Stalk variants showed higher affinity than the shorter NKp30_LBD construct when analyzed by SPR (Figure 7c), thus confirming earlier observations that the stalk region contributes significantly to B7-H6 binding. The affinity of the NKp30_Stalk monomeric fraction was approximately two times higher than that of NKp30_LBD, as shown by both SPR and ITC analyses (Figure 7b,d). Even higher affinities were recorded by SPR for the NKp30_Stalk oligomeric fraction, which was bound approximately ten times more strongly than NKp30_LBD and four times more strongly than the NKp30_Stalk monomeric fraction. Simultaneously, the *B_max_* was higher, as expected for oligomer binding (Figure 7c) and in line with a previous study, thus jointly concluding that the observed increase in affinity results from the avidity contribution of the oligomers [21].

Surprisingly, the affinity of the NKp30:B7-H6 interaction was similar when comparing NKp30_LBD with the monomeric and oligomeric fractions of NKp30_Stalk by ITC (Figure 7d,e), albeit with a tenfold decrease in affinity of the oligomeric fraction compared to SPR results. In addition, ITC results showed lower stoichiometry for the oligomeric fractions of NKp30_Stalk (Figure 7f), thus suggesting that not all binding sites of the oligomer are saturated and that they might be sterically blocked and inaccessible for B7-H6 binding. These findings match our results from the sedimentation analysis (Figure 6c), i.e., only 60–70% NKp30_Stalk T oligomers are accessible.

### 2.6. Crystal Structure of Glycosylated NKp30:B7-H6 Complex

To understand the role of glycosylation in the NKp30:B7-H6 interaction in more detail, we solved the crystal structure of NKp30 with uniform glycosylation in complex with Endo F1-deglycosylated B7-H6. Although we were unable to crystallize this complex with the oligomer-forming NKp30_Stalk construct, we obtained diffracting crystals using the NKp30_LBD monomeric protein. The structure (PDB 6YJP) was solved at 3.1 Å resolution, and the refinement parameters are summarized in Table 1. The asymmetric unit of the crystal contains two NKp30_LBD molecules (LBD_A/B) and three B7-H6 molecules (B7-H6_C/D/E), as shown in Figure 8 below. The molecules are completely localized, except for two missing loops in B7-H6 (residues 151–159 and 149–155 in chains D and E, respectively). In fact, one more NKp30_LBD chain that interacts with B7-H6 (chain E) is present in the crystal but its loose localization did not allow us to build it in the structure.

When viewing neighboring symmetry-related molecules, surrounding the asymmetric unit of the crystal, a dimer of two NKp30_LBD:B7-H6 complexes (LBD_A:B7-H6_C and LBD_B:B7-H6_D) becomes apparent (Figure 9). The interaction interfaces between NKp30 and B7-H6 are conserved and highly similar to the interaction surface observed in the 3PV6 structure; the contact between NKp30_LBD_A and B7-H6_C chains comprises four hydrogen bonds or salt bridges (chain A Gly51 N—chain C Thr127 OG1, chain A Val53 N—chain C Pro128 O, chain A Glu111 OE1—chain C Lys130 NZ and chain A Glu111 OE2—chain C Lys130 NZ), and the interface has a Pisa server complex formation significance score of 0.225 [27]. The contact between chains NKp30_LBD_B and the symmetry-related B7-H6_D comprises five hydrogen bonds or salt bridges: four analogical to the aforementioned bonds and an additionally B47 Arg NH1—D84 Asp OD1 bond. The interface has the same score, 0.225, according to the Pisa server.

In the crystal structure, electron density maps allowed us to model *N*-acetylglucosamine at NKp30_LBD asparagine residues 42 (chains A and B), and B7-H6 residues 208 (chains C and D) and 43 (chain E). Furthermore, low-quality peaks in the electron density map, corresponding to glycans, were observed near NKp30_LBD asparagine residues 68 (chain A) and 121 (chain B), and near B7-H6 residues 43 (chains C,D), 57 (chains C,D,E), 174 (chains C,D,E), 208 (chain E), and 242 (chains C,E). In the case of NKp30_LBD Asn42, the glycosylation changes the orientation of the Asn42 side chain and main chain and consequently, the placement of residue Ala43 in comparison with NKp30_LBD in PDB 3NOI and 3PV6. Noticeably, the glycosylation site at Asn42 of one NKp30_LBD molecule (LBD_A) localizes in the proximity of the C-terminus of the other NKp30_LBD molecule (LBD_B) and vice versa (Figure 9). In NKp30_Stalk or in full-length NKp30 receptor, this C-terminal part of its ligand binding domain would be followed by the 15 amino acids long stalk region. Both the glycosylation and the stalk regions are required for NKp30 oligomerization, suggesting that this NKp30_LBD dimer may be the building block of these oligomers.

NKp30_LBD chains A and B form a dimer different from that observed in the crystal structure of NKp30 itself (PDB 3NOI) [19], whereas NKp30 was only monomeric in the crystal structure of the NKp30:B7-H6 complex (PDB 3PV6) [20]. Both dimers have two-fold symmetry and a similar patch of mutual contacts (residues Arg28, Asn42, Gln45, and Glu128 participate in hydrogen bonds in both cases); however, their mutual orientation is very different: when chains A are superimposed, the positions of the same residues in chain B differ by 9–15 Å (Figure 10). The NKp30 dimer in PDB 3NOI comprises sixteen hydrogen bonds or salt bridges, and the interface scores 0.176 in the Pisa server, whereas the dimer observed in the present structure PDB 6YJP comprises eleven hydrogen bonds or salt bridges, and its interface scores 1.0 in the Pisa score, which indicates a more stable interaction and relevant interface. Interestingly, Asn42 of NKp30_LBD is positioned directly at the dimer interface in PDB 3NOI, and in close contact with Glu26 and Arg28 from the second chain, whereas GlcNAc at Asn42 in PDB 6YJP is located outside of the interface, right next to the C-terminus of the second chain—at the beginning of the stalk region.

## 3. Discussion

The first key stage of this study was the preparation of the stable, soluble, extracellular part of the B7-H6 protein. Members of the B7 family have moved to the forefront of cancer research for their underlying involvement in tumorigenesis [17] and tumor recognition [16] and for their role as regulators of immune responses and immunotherapy outcomes, such as B7-H1, also known as PD-L1, one of the most discussed checkpoint inhibitors in recent years [28]. Because B7-H6 can induce NKp30-dependent NK activation and cytokine secretion [12], therapeutic interventions based on the NKp30:B7-H6 interaction may provide a new strategy for tumor treatment. Wu et al. [29] have shown that the B7-H6-specific bispecific T cell engager (BiTE) directs host T cells to mediate cellular cytotoxicity and interferon-γ secretion, which is therefore a potential therapeutic strategy for B7-H6^+^ hematological and solid tumors. More recently, T cells expressing B7-H6-specific human single-chain fragment variable (scFv) as chimeric antigen receptor (CAR) have been shown to induce potent anti-tumor activity in vitro and in vivo against tumors expressing high amounts of B7-H6 [30].

Although B7-H6 was the target molecule in these approaches, other strategies have been developed exploring the potential of B7-H6 as a natural ligand for the activating receptor NKp30. Kellner et al. [31,32] generated a fusion protein consisting of the ectodomain of B7-H6 and of the CD20-specific scFv 7D8. In the functional assay, the authors found that the B7-H6:7D8 fusion protein could stimulate NKp30-mediated NK cell cytotoxicity. The same strategy was successfully applied to create a HER2-specific B7-H6 fusion protein targeting HER2^+^ tumors [33]. In our study, we initially tried to express not only the B7-H6 construct described above (Figure 1), but also a shorter construct corresponding to the N-terminal IgV B7-H6 domain (Asp25—Val140) only. However, this shorter construct could not be expressed at all, thus suggesting that, although the two Ig-like domains of B7-H6 are structurally well separated, the extracellular part of B7-H6 is stable only when expressed as a whole. Furthermore, we found that mutating the odd cysteine in the B7-H6 IgC domain (Cys212Ser) greatly stabilizes the molecule by promoting correct disulfide bond formation, reaching a ten-fold increase in yield over the wild-type B7-H6 ectodomain. Moreover, glycerol addition was optimal for long-term B7-H6 storage, both in solution and in frozen state, preventing its aggregation, especially at higher protein concentrations. This optimized protocol for recombinant B7-H6 production may be useful for further studies involving B7-H6 fusion immunotherapeutics or requiring large-scale expression of this tumor antigen in a stable and well-folded form.

Oligomerization of the NKp30 ectodomain has been previously characterized using constructs expressed in Sf9 insect cells providing uniform, simple paucimannose *N*-glycans, similar to those present in HEK293S GnTI^−^ cell lines. When recombinantly expressing these constructs, the authors found that both NKp30_Stalk and NKp30_LBD oligomerized in their study, although the latter to a lesser extent [21]. However, both proteins had a considerable number of additional amino acid residues at both their N- and C-termini, namely ADLGS and GSENLYFQGGS followed by a decahistidine tag, respectively. In our study, we used the same LBD and Stalk regions of the NKp30 ectodomain, but we expressed these NKp30 constructs in human cell lines, providing uniform mannose- or wild-type glycosylation, and with a limited number of flanking amino acids. Regardless of the glycosylation type or presence, at low concentration the NKp30_LBD construct showed no tendency to oligomerize. At high sample concentration, however, dimer formation has been observed for this construct in our sedimentation analysis (Appendix A). Thus, we can conclude that although the ligand binding domain of NKp30 itself shows some tendency to self-associate (as evidenced by dimer formation observed at high concentration in solution by AUC and by dimeric arrangement found in the crystal), presence of the stalk region is a prerequisite for stable NKp30 oligomerization in solution.

In addition, the ability of the NKp30_Stalk construct to oligomerize is lost upon its deglycosylation with endoglycosidase F1, thereby highlighting the unappreciated but key role of glycosylation in NKp30 oligomerization. Hermann et al. [21] observed that NKp30_Stalk oligomers bound to immobilized B7-H6-Ig have an extremely low nanomolar *K_D_*, as assessed by SPR and ELISA (both surface-based interaction methods), and hypothesized that the increase in apparent ligand binding affinity of the oligomers is caused by the increase in the avidity of higher-molecular-order NKp30 complexes under these conditions. We noted a similar trend in our SPR analysis when using immobilized B7-H6, observing a higher affinity for NKp30_Stalk than for NKp30_LBD, and an even higher affinity for the NKp30_Stalk oligomeric fraction. In contrast, when analyzing the same system by AUC and ITC (solution-based techniques), we found that both monomeric and oligomeric fractions of NKp30_Stalk, and NKp30_LBD, showed similar thermodynamic parameters and affinity to soluble B7-H6, but the oligomers exhibited significantly lower binding stoichiometry. This suggests that interaction data on immobilized B7-H6 and oligomerizing NKp30_Stalk construct collected using surface methods might not correctly express *K_D_* values for interactions of individual binding sites and that the NKp30 oligomers might not be completely biologically active species, at least when present in solubilized form and not on the cell surface.

Another important aspect of the NKp30:B7-H6 interaction is the glycosylation of the NKp30 ectodomain, especially on Asn42 and Asn68 sites, as shown by Hartmann et al. [22] in binding experiments with NKp30 human IgG1-Fc fusion constructs expressed in HEK293T cell lines. Interestingly, mutating these glycosylation sites affected binding to B7-H6, but not to BAG-6, as determined by SPR or ELISA, respectively. Moreover, NKp30_Stalk-Ig binding to natural ligands, in various tumor cell lines, was even stronger in the absence of glycosylation, albeit abrogated in B7-H6^+^ reporter cell lines. This key role of glycosylation and of the stalk region in signal transduction was further corroborated by using various NKp30-transfected CD3ζ reporter cell lines stimulated by B7-H6^+^ target cells [22]. In our study, we did not observe marked differences in NKp30:B7-H6 affinity between the NKp30_Stalk and NKp30_LBD constructs, and neither between the NKp30 variants expressed with wild-type human or uniform oligomannose *N*-glycans, or deglycosylated, independent of the method that we used to measure these interactions. The deglycosylated proteins used in our study still had a single GlcNAc residue at each glycosylation site, in contrast to the disruptive Asn-Gln mutations used in the previous study, which may account for the differences in the results. Nevertheless, our findings indicate that glycosylation most likely does not directly affect the NKp30:B7-H6 interaction and instead primarily affects the ability of NKp30 to oligomerize. Impaired oligomerization would then translate into lower apparent binding affinities, when using surface-based methods such as SPR and ELISA, and into impaired signal transduction in cell-based experiments. The role of the stalk domain in CD3ζ-mediated activation of NK cells was thoroughly characterized by Memmer et al. [7] who showed that mutations in the stalk region close to LBD weaken the *K_D_* of B7-H6-Fc whereas BAG-6 binding, again, remained mostly unaffected, although NKp30-Fc IgG fusions and SPR detection were used in these experiments. Subsequent reporter cell-based assays showed that Arg143 (end of the NKp30 stalk region) alignment with the aspartate of CD3ζ is required for signal transduction and that this alignment might be achieved by ligand-induced receptor clustering and/or stalk-dependent conformational changes [7]. NKp46, another member of the NCR family, exhibits similar complex behavior on the cell surface. It forms dimers and later on also clusters within the immune synapse, which activate NK cell polarization [34,35].

Although NKp30 constructs artificially dimerized through IgG-Fc fusion may not be the best tools to describe the natural behavior of this receptor at the plasma membrane of NK cells, their thorough characterization should be useful for developing immunotherapeutics. Over ten years ago, Arnon et al. [36] demonstrated using human prostate cancer cell lines that treatment with NKp30-Ig dramatically inhibits tumor growth in vivo in mice by successfully recruiting activated macrophages via antibody-dependent cellular cytotoxicity (ADCC). Strikingly, while IgG1 Fc-fusions are regularly used in NCR-related immunology research, no immunotherapy product based on them has been developed thus far.

The original motivation for our work was to understand better how NKp30 oligomers are formed and structured. To this end, we used multiple techniques, but not all of them produced conclusive results, such as structural mass spectrometry (cross-linking and H/D exchange) or cryo-electron microscopy. We also tried to crystallize the NKp30_Stalk construct and thus solve its oligomeric structure, albeit to no avail. Neither this protein nor its complex with B7-H6 formed crystals. Nevertheless, we solved the structure of the glycosylated NKp30_LBD:B7-H6 complex, which is somewhat similar to the previously known structures of NKp30_LBD (PDB 3NOI, [19]) and of the NKp30_LBD:B7-H6 complex (PDB 3PV6, [20]). However, both previously published structures were solved using bacterially expressed, non-glycosylated refolded NKp30_LBD. Therefore, our structure provides new insights into the mechanism of oligomerization of this protein.

Most interestingly, our crystal structure shows the formation of an NKp30_LBD dimer in the bound state with B7-H6. The symmetrical arrangement of Asn42 glycosylation sites, near the C-terminal stalk region beginnings on both sides of the dimer, strengthens our hypothesis of glycosylation-supported, stalk region-mediated NKp30 oligomerization. In contrast to the structure of the glycosylated NKp30_LBD dimer, in the dimer of the refolded NKp30_LBD observed in the PDB 3NOI crystal structure, the Asn42 residue is in close contact with neighboring side chains of other amino acids of the dimerization interface. With such a bulky glycan chain bound to the Asn42, this type of dimer is unlikely to exist. Therefore, we may assume that glycosylation induces the observed dimer arrangement and that this arrangement is further stabilized by the interaction between the glycan moiety and the stalk region of full-length NKp30.

Moreover, for soluble NKp30_Stalk, the stalk regions that extend on both sides of the dimer would be free to interact with stalk regions of other NKp30_Stalk dimers, thereby forming linear oligomers composed of such dimers. To acquire at least some low-resolution data on the structure of these oligomers, we have also performed SEC-SAXS analysis of the NKp30_LBD, NKp30_Stalk monomeric and oligomeric fractions, and their complexes with B7-H6 (Appendix A). Representative examples of the resultant ab initio molecular envelopes calculated from the collected SAXS data are shown in Appendix A [37,38]. The lack of further structural data on how individual NKp30_Stalk molecules or their possible dimers are arranged in space precludes any further modelling of molecular arrangements of these oligomers. Nevertheless, the overall shape of the calculated envelopes is highly asymmetrical, that is, prolate or oblate rather than spherical, and this asymmetry is in perfect agreement not only with the prediction based on the crystal structure but also with our sedimentation analysis of the oligomers indicating asymmetrical elongated or flattened particles.

Another interesting aspect of our crystal structure is the overall topology of the dimer of the NKp30_LBD:B7-H6 complex observed in the crystal lattice. Its arrangement is compatible with both NKp30 molecules inserted within the same NK cell membrane and B7-H6 in the cell membrane of a target tumor cell (Figure 11). Such arrangement would bring the membranes of both cells into close contact, and such an effect could be further potentiated by NKp30 oligomerization. Importantly, Xu et al. [39,40] analyzed the crystal structure of the Fab of inhibitory antibody 17B1.3 in complex with the ectodomain of B7-H6 and found that 17B1.3 could bind to a site on B7-H6 that was completely different from the binding site on NKp30 (PDB 4ZSO). Using an NKp30 reporter cell line and B7-H6-expressing P815 tumor cells, they concluded that the bulky 17B1.3 antibody acts by sterically interfering with close cell–cell contacts at the NK cell–target cell interface, thereby blocking immunological synapse formation and NK cell activation [40].

These results support the proposed close interaction mediated by the NKp30 dimer ligated with two B7-H6 monomers, as observed in the presented crystal structure. However, further studies are required to confirm such an arrangement directly on the membrane of living NK cells, as well as the oligomeric arrangement of NKp30. Considering the short length of the stalk region, it is unlikely that NKp30 forms octameric, decameric, or even larger oligomers through its stalk region on the cell surface, in contrast to its oligomerization in solution. Therefore, data on NKp30 receptor oligomers collected in solution, using its solubilized ectodomains interacting in three-dimensional (3D) space, should not be directly extrapolated to its natural habitat within the plasma membrane and to a 2.5D space on the cell surface. Moreover, all future immunotherapeutic strategies and reagents designed to target or trigger this NK cell receptor:ligand system should also allow enough flexibility with respect to the B7-H6 moiety, thus ensuring its proper orientation and interaction with NKp30.

## 4. Materials and Methods

### 4.1. Cell Culture and Vector Design

HEK293T cells were kindly provided by Radu A. Aricescu [24]. HEK293T and HEK293S GnTI^−^ cells [25] (0.3–6 × 10^6^/mL) were maintained in ExCELL293 serum-free medium (Sigma, St. Louis, MT, USA) supplemented with 4 mM L-glutamine in square-bottom glass DURAN flasks with gas permeable caps (DWK Life Sciences, Wertheim, Germany) using 30–40% of the nominal volume at 135 rpm on orbital shaker placed within the incubator at 37 °C and 5% CO_2_ [26]. Transient expression in both HEK293T and HEK293S GnTI^−^ cell lines was performed using the plasmid pTW5sec, a derivative of the pTT5 plasmid backbone [41] modified in-house to contain a woodchuck hepatitis virus post-transcriptional regulatory element (WPRE) and a leader peptide of secreted alkaline phosphatase in frame with AgeI and KpnI cloning sites, followed by a C-terminal histidine tag. Therefore, proteins recombinantly expressed using this plasmid were secreted into the cell culture media and purified by immobilized metal affinity chromatography (IMAC).

### 4.2. Protein Expression and Purification

For high-density transfection, 800 × 10^6^ HEK293 cells were centrifuged for 5 min at 90× *g*. Cells were resuspended in 34 mL of ExCELL293 medium in an empty flask and 800 μg of DNA (1 μg/10^6^ cells) were diluted in 6 mL of phosphate-buffered saline (PBS) and filtered through a 0.22 μm filter into the flask with cells. Subsequently, linear 25 kDa polyethyleneimine (lPEI) was added in a 1:3 DNA:lPEI (*w*/*w*) ratio. Cells were incubated in high density on a shaker for 1.5–4 h at 37 °C. Afterwards, up to 400 mL ExCELL293 medium and 0.5 M valproic acid [42] were added to a final concentration of 2 mM. Cells were harvested after 7 days or earlier, when viability dropped below 70%, by centrifugation for 30 min at 15000× *g* at 20 °C. The supernatant was filtered through a 0.2 μm filter, diluted two-fold with PBS buffer, and loaded onto a pre-equilibrated 5 mL HiTrap TALON column using an ÄKTAprime FPLC system (GE Healthcare, Chicago, IL, USA). Protein was eluted with 250 mM imidazole in PBS, concentrated using Amicon Ultra (MWCO 10000; Sigma, St. Louis, MT, USA) concentrators, filtered using spin filter and subjected to size-exclusion chromatography on Superdex 200 Increase 10/300 GL column (GE Healthcare, USA) as a final purification step with HEPES buffer (10 mM HEPES pH 7.5, 150 mM NaCl, 10 mM NaN_3_) as the mobile phase. For NKp30_Stalk, fractions were collected separately for oligomeric and monomeric species, and immediately frozen in liquid nitrogen. Protein glycosylation and disulfide bonds pairing were characterized by mass spectrometry (for details see Appendix A) [43,44,45].

### 4.3. Protein Labelling

B7-H6 expressed in HEK293S GnTI^−^ cells was stained with ATTO 488 fluorescent dye (Sigma, USA) using NHS labelling chemistry according to the manufacturer’s instructions. Nine volumes of B7-H6 solution in HEPES buffer were mixed with one volume of 1 M bicarbonate buffer, pH 8.5. ATTO 488-NHS ester was added to the protein solution at a molar ratio of 1:3 (protein:ATTO488-NHS). Reaction mixture was incubated for 60 min in the dark at room temperature on a roller shaker at 35 rpm. The labelled protein was purified on a HiTrap Desalting column (GE Healthcare, USA) connected to an ÄKTA basic HPLC system.

### 4.4. Deglycosylation

For deglycosylation of NKp30 and B7-H6 expressed in HEK293S GnTI^−^ cell lines, the proteins in HEPES buffer at 1 mg/mL concentration were mixed with recombinant GST-tagged endoglycosidase F1 (Endo F1) [46] in a 200:1 (*w*/*w*) ratio (target protein:Endo F1). For B7-H6, the buffer was supplemented with 20% glycerol to increase protein stability. The mixture was incubated for 3 h at 37 °C or at 4 °C overnight while mixing. Then, the solution was loaded onto a 5 mL GST-trap column (GE Healthcare, Chicago, IL, USA) connected to the fast protein liquid chromatography (FPLC) system to capture the Endo F1, while collecting the flow-through containing deglycosylated protein. The solution was concentrated to 200 µL and loaded onto a Superdex 200 Increase 10/300 GL (GE Healthcare, Chicago, IL, USA) column for further purification and characterization of the deglycosylated proteins. Samples from collected fractions were used for SDS-PAGE analysis.

### 4.5. Differential Scanning Fluorimetry

Differential scanning fluorimetry was performed using SYPRO Orange Protein Gel Stain (Thermo Scientific, Waltham, MA, USA). Its 5000× DMSO stock solution was diluted to 50× concentrated solution. The samples were prepared by mixing 4 μL of B7-H6 expressed in HEK293S GnTI^−^ cells with 10 μL of 2× concentrated screen buffer and 1 μL of 50× concentrated stain solution and complemented by HEPES buffer up to 20 μL. The sample without protein was used as a blank, and the sample with no additive was used as a reference. Data were collected on a Rotor-Gene 2000 Real Time Cycler (Corbett Research, Sydney, Australia) at an excitation wavelength of 300 nm. Fluorescence was measured at 570 nm. The temperature increased stepwise, 0.5 °C each 30 s from 25 °C to 95 °C. The data were evaluated using OriginPro 8 software (version 8.500161). Melting points can be extrapolated by finding the inflection point of sigmoid melting curves. By depicting the data as a negative derivative of relative fluorescence (−d(RF)/T), the melting points correspond to the minima of the melting curves.

### 4.6. Sedimentation Analysis

Sedimentation velocity measurements were performed on an analytical ultracentrifuge ProteomeLab XL-I (Beckman Coulter, Brea, CA, USA). Protein samples were purified by gel filtration, using the mobile phase buffer (10 mM HEPES pH 7.5, 150 mM NaCl, 10 mM NaN_3_) as a reference. Sedimentation velocity experiments were conducted with 20 μM protein samples using double sector cells and An50-Ti rotor at 20 °C and 36,000 or 48,000 rpm for oligomeric or monomeric samples, respectively. Absorbance scans were recorded at 280 or 480 nm at 3–7 min intervals. Buffer density, protein partial specific volume, and particle dimensions were estimated in Sednterp (www.jphilo.mailway.com). Data were analyzed in Sedfit [47] using the continuous sedimentation coefficient distribution c(s) model. Analysis of NKp30_LBD at high concentration was performed with its 20 mg/mL sample at 50,000 rpm using 3 mm centerpiece; scans were recorded every 2 min using the interference optics. The data were fitted with the nonideal c(s) model in Sedfit [48].

### 4.7. Size-Exclusion Chromatography with Multi-Angle Laser Light Scattering (SEC-MALS)

SEC-MALS experiments were performed using an FPLC station equipped with miniDAWN Tristar light scattering (Wyatt Technologies, Santa Barbara, CA, USA) and Shodex RI-101 (Showa Denko K.K., Tokyo, Japan) refractive index detectors with a Superose 6 Increase 10/300 GL column (GE Healthcare, Chicago, IL, USA) equilibrated in 20 mM HEPES pH 7.5, 150 mM NaCl, 10 mM NaN_3_ buffer. Single proteins or protein complexes, typically at 1 mg/mL concentration, were loaded onto the column. The molecular weight was estimated using the refractive index as measure of concentration. The results were analyzed using Astra software (Wyatt Technologies, Santa Barbara, CA, USA).

### 4.8. Isothermal Titration Calorimetry

Thermodynamic parameters of NKp30:B7-H6 interactions were determined using the PEAQ-ITC instrument (Malvern Panalytical, Westborough, MA, USA). All measurements were performed in the 10 mM HEPES pH 7.5, 150 mM NaCl, 10 mM NaN_3_ buffer. First, control heat was determined by buffer–buffer titration. For protein interaction measurements, 200 µL of 20–30 µM NKp30 variants were loaded into the cell and 40 μL of 200–300 µM B7-H6 into the syringe (these concentrations varied slightly in different experiments). An initial injection of 0.4 µL of B7-H6 was followed by 25 injections of 1.5 µL. Injection duration was 2 s, in 120 s intervals, performing the measurements at 25 °C and stirring the cell solution at 750 rpm. The data were evaluated using NITPIC [49], Sedphat [50], and GUSSI software [51].

### 4.9. Surface Plasmon Resonance

Surface plasmon resonance (SPR) experiments were performed to measure NKp30_Stalk and NKp30_LBD binding, in three different *N*-glycosylation states (wild-type glycans, simple glycans, and deglycosylated with Endo F1), to B7-H6, also in three glycosylation states. Measurements were performed using the Biacore T200 system (GE Healthcare, Chicago, IL, USA) in a buffer composed of 10 mM HEPES pH 7.5, 150 mM NaCl, 1 mg/mL dextran, and 0.05% Tween-20. Three glycosylation variants of B7-H6 were biotinylated (EZ-Link NHS-Biotin; Thermo Fisher Scientific, Waltham, MA, USA) according to the manufacturer’s instructions and immobilized to the streptavidin sensor. Binding experiments were performed in single cycle kinetics mode with 15 sequential injections of NKp30 samples in each cycle with concentrations ranging from 1 nM to 16.3 μM. Data from a reference flow cell with an empty channel were subtracted and fitted using the Biacore T200 evaluation software (version 3.0). Maximal fitted responses from each cycle were analyzed using Sedphat [50] software using the AB hetero-association model.

### 4.10. Protein Crystallization

Protein crystallization was performed at the Division of Structural Biology of the Wellcome Centre for Human Genetics, University of Oxford. Initial screening was performed using the sitting drop method in 300 nL (100 nL of protein solution and 200 nL of crystallization reagents) using the Hydra and Cartesian instruments. All crystallization plates were stored in Rock Imager (Formulatrix, Bedford, MA, USA) at 21 °C. Four commercially available screens (Index, Proplex, PACTpremier, Crystal screen) were used for the equimolar mixture of deglycosylated B7-H6 with NKp30_Stalk or NKp30_LBD, both expressed in HEK293S GnTI^−^ cells; the mixture was concentrated to 10 mg/mL before the drop set-up. Initially, needle-shaped crystals of NKp30_LBD:B7-H6 complex were obtained in 0.1 M sodium citrate pH 5.0, 20% PEG 8000. These crystals were crushed and used for seeding in optimization performed using the sitting drop method in 300 nL (200 nL of protein solution and 100 nL of crystallization reagents). The number of crystals grew in the drops that were seeded, and diffraction data were collected for a few of them. 25% glycerol was used as cryoprotectant and data were collected at 100 K. The best diffraction data were collected from crystals grown in 0.1 M sodium citrate pH 6.7, 11.7% PEG 6000.

### 4.11. Diffraction Data Collection

Eleven diffraction data sets were collected at the Diamond Light Source (Didcot, Oxfordshire, UK) at beamline I02 using a wavelength of 0.97949 Å and a PILATUS 6M-F detector (Dectris, Baden-Daettwil, Switzerland). The resolution of all data sets was 3–4 Å and the crystals degraded during the data collection. Eventually, three data sets from three crystals (set B7x1: images 1 to 800 (80°), set B8x5:1 to 800 (80°), and set B8x3:1 to 450 (45°)) were merged to process the data for optimal results. The data were integrated in XDS [52] and merged and scaled in AIMLESS from the CCP4 software package [53], in space group *C*2 (recommended by POINTLESS and ZANUDA) and alternatively in P1. Finally, *C*2 was selected as the correct space group based on the phase problem solution and refinement. Data suffered by strong anisotropy, with effective resolution in the direction of axis *a* being the lowest, ca. 4.4 Å. Anisotropy corrections were not applied to the data. Final data processing statistics are shown in Table 1.

### 4.12. Structure Solution and Refinement

The phase problem was solved in space group *C*2, using MORDA [54] and PHASER [55], data cut off 3.8 Å and already known structures of NKp30 and B7-H6 as models (PDB code 3PV6 for both molecules). The asymmetric unit in *C*2 contains three B7-H6 protein chains and two NKp30 chains. After refinement trials in REFMAC5 [56], considering manual optimization of restraints, B-values and translation, rotation, and screw-rotation (TLS) parameterization, the structures were refined in LORESTR (automatic REFMAC5 pipeline for low-resolution structure refinement, [57]), using atomic B-factors, without TLS and with restraints to homologs optimized specially for each run of the refinement. Five percent of reflections were used as a test set (*R*_free_ set). Manual editing was performed in COOT [58]. Structure quality was checked using the validation tools implemented in MOLPROBITY [59]. There are four Ramachandran outliers (chain C Gly83 and Val152, chain D Gly83, and chain E Gln182), which is 0.5% of refined residues. The final structure parameters are outlined in Table 1.

### 4.13. Data and Structure Deposition

Diffraction data have been deposited in the SBGrid Data Bank under code 753 (doi:10.15785/SBGRID/753). The crystal structure has been deposited in the Protein Data Bank under code 6YJP.

## 5. Conclusions

Glycosylation is an essential post-translational modification of cell surface proteins, not only stabilizing them in the extracellular environment but also providing new functional modalities. Moreover, glycosylation is crucial for immune recognition and self-non-self discrimination, especially when involving receptors of the innate immune system, which often bind to their ligands with weak affinities. Accordingly, their ability to transduce the signal into the cell should significantly vary as a function of their glycosylation presence and its type. Glycans may help these receptors to create stable signaling complexes through dimerization and oligomerization or to organize them within complex dynamic structures of the immune synapse. In this study, we exemplified this behavior for the NK cell activation receptor NKp30, whose oligomerization depends on its *N*-glycosylation. Furthermore, we solved the first crystal structure of a glycosylated NKp30 ligand binding domain, in a complex with its tumor ligand B7-H6, highlighting why glycosylation is crucial for the NKp30 oligomerization and for signal transduction. Furthermore, our results indicate that glycosylation should not be overlooked when planning or conducting structure-function studies. In conclusion, our structure of the complex between glycosylated NKp30 and B7-H6 provides a template for designing molecules to stimulate NKp30-mediated cytolytic activity for tumor immunotherapy.

## Figures and Tables

**Figure 1 cancers-12-01998-f001:**
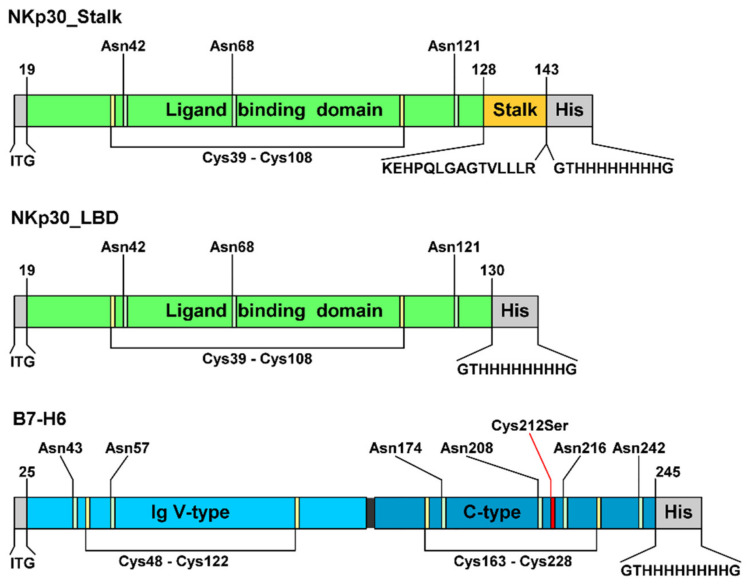
Recombinant NKp30 and B7-H6 expression constructs. All constructs contain three amino acids which remain after the secretion signal is cleaved at the N-terminus and a histidine tag sequence at the C-terminus. Glycosylated asparagine residues and cysteines forming disulfide bridges are indicated, as well as the mutation of the odd cysteine C212S in B7-H6.

**Figure 2 cancers-12-01998-f002:**
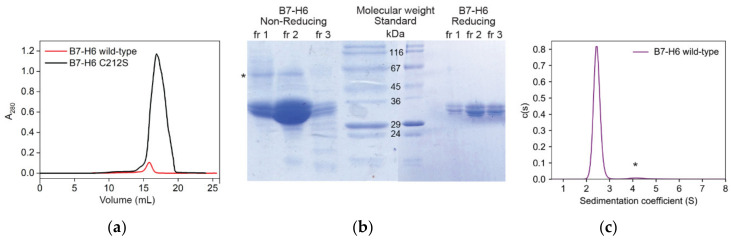
Recombinant B7-H6 is stabilized by the C212S mutation. (**a**) Size-exclusion chromatography (SEC) profiles of recombinantly expressed wild-type B7-H6 and its C212S mutant. (**b**) Fractions of the wild-type B7-H6 SEC peak were analyzed by 15% SDS-PAGE under non-reducing (left) and reducing (right) conditions. (**c**) Sedimentation analysis of wild-type B7-H6 at 0.5 mg/mL shown as continuous size distribution of the sedimenting species c(s). Dimeric species are marked with an asterisk in (**b**,**c**).

**Figure 3 cancers-12-01998-f003:**
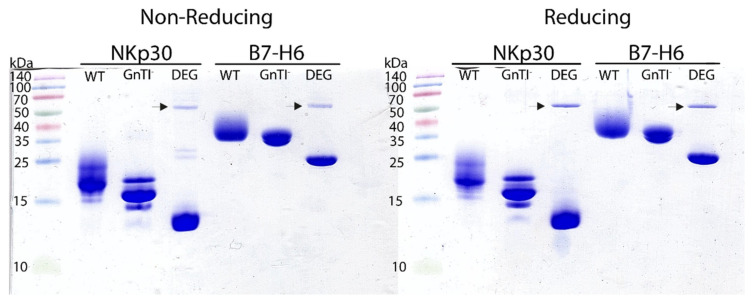
SDS-PAGE analysis of deglycosylated NKp30_LBD and B7-H6 C212S mutant. Different glycosylation states are marked as WT for proteins expressed in HEK293T cells with wild-type (WT) *N*-glycosylation; GnTI^−^ expressed in HEK293S GnTI^−^ cells lacking *N*-acetylglucosaminyltransferase I activity which have, therefore, uniform Asn-GlcNAc_2_Man_5_
*N*-glycans; and DEG for the GnTI^−^ proteins deglycosylated (DEG) with endoglycosidase Endo F1 (itself marked by an arrow).

**Figure 4 cancers-12-01998-f004:**
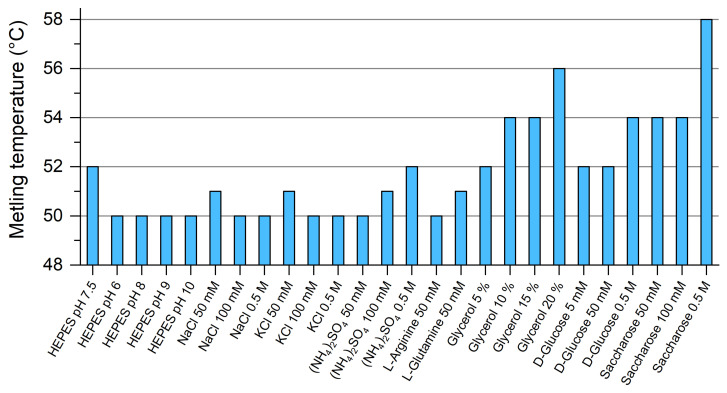
B7-H6 is stabilized by glycerol and saccharose addition. Differential scanning fluorimetry was used to analyze changes in protein melting temperature when adding various reagents. The concentrations given in the graph correspond to the final concentrations present in the sample.

**Figure 5 cancers-12-01998-f005:**
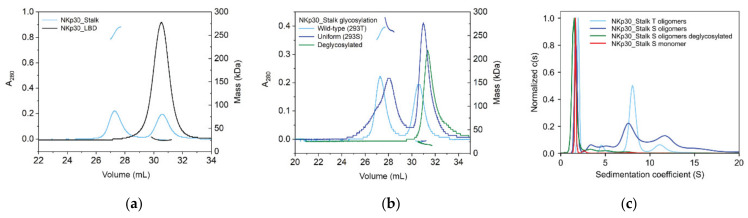
Glycosylation is necessary for NKp30 oligomerization. (**a**) SEC-MALS analysis of NKp30_Stalk and NKp30_LBD, confirming that the stalk region is required for oligomerization. (**b**) SEC-MALS analysis of recombinantly expressed NKp30_Stalk with wild-type glycosylation (black), uniform, simple glycans (blue), both showing non-covalent oligomers, and deglycosylated sample, which does not form oligomers (red). (**c**) Normalized continuous size distributions of sedimenting species for glycosylated and deglycosylated NKp30_Stalk oligomers and for its monomeric fraction. The main peak corresponds to the NKp30_Stalk monomer, whereas a broad distribution of oligomeric species is present in glycosylated NKp30_Stalk samples.

**Figure 6 cancers-12-01998-f006:**
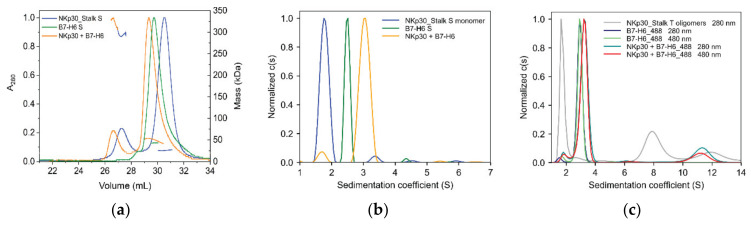
NKp30_Stalk oligomers cannot be saturated by B7-H6. (**a**) Normalized SEC (Size Exclusion Chromatography) elution profiles of NKp30_Stalk, B7-H6 and their equimolar mixture with MALS detection. (**b**) Sedimentation analysis of complex formation for B7-H6 and NKp30_Stalk monomeric fraction and (**c**) of NKp30_Stalk oligomeric fraction with B7-H6 labeled with ATTO488 dye performed at two wavelengths. Letters S and T denote the type of protein glycosylation (HEK293S GnTI^−^ or HEK293T *N*-glycans, respectively).

**Figure 7 cancers-12-01998-f007:**
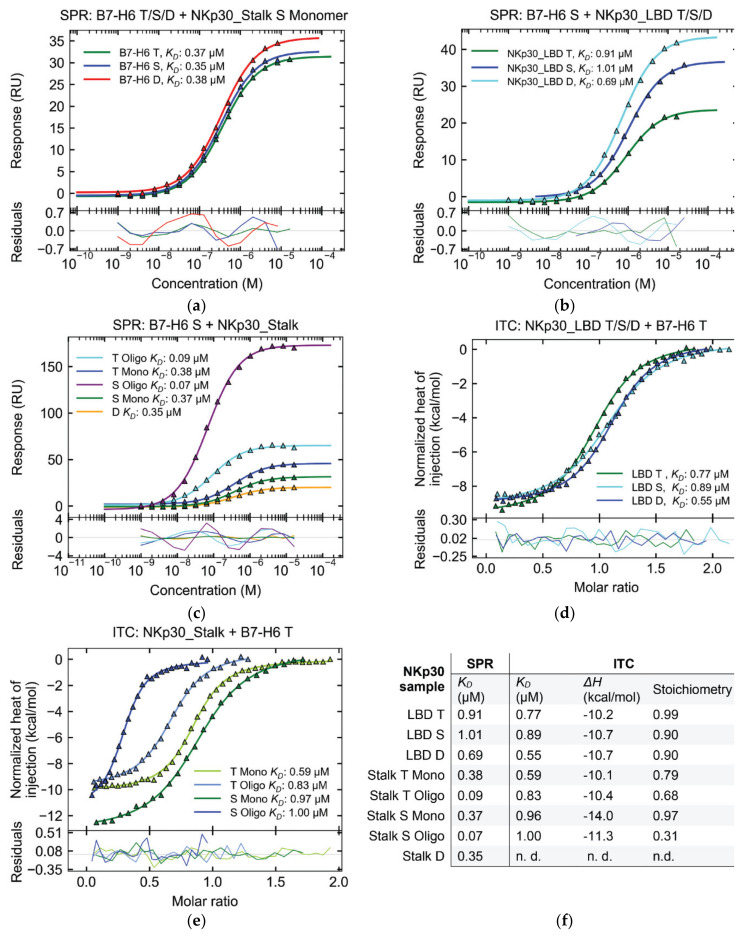
Characterization of the NKp30:B7-H6 interaction by SPR and ITC. Type of protein glycosylation: T—wild-type, S—uniform, D—deglycosylated, Mono—monomeric, Oligo—oligomeric fraction of NKp30_Stalk. Titles above individual graphs describe the given experiment: SPR or ITC, protein bound to SPR sensor or present in ITC cell + protein flown over the sensor or titrated into the cell, respectively. (**a**) B7-H6 glycosylation does not affect NKp30 binding. (**b**,**d**) Glycosylation of NKp30 weakens its interaction with B7-H6. (**c**,**e**) Stalk region and oligomerization of NKp30 enhance its affinity to B7-H6 in SPR but not in ITC. (**f**) Comparison of all thermodynamic parameters measured for the NKp30:B7-H6 interaction with different protein variants.

**Figure 8 cancers-12-01998-f008:**
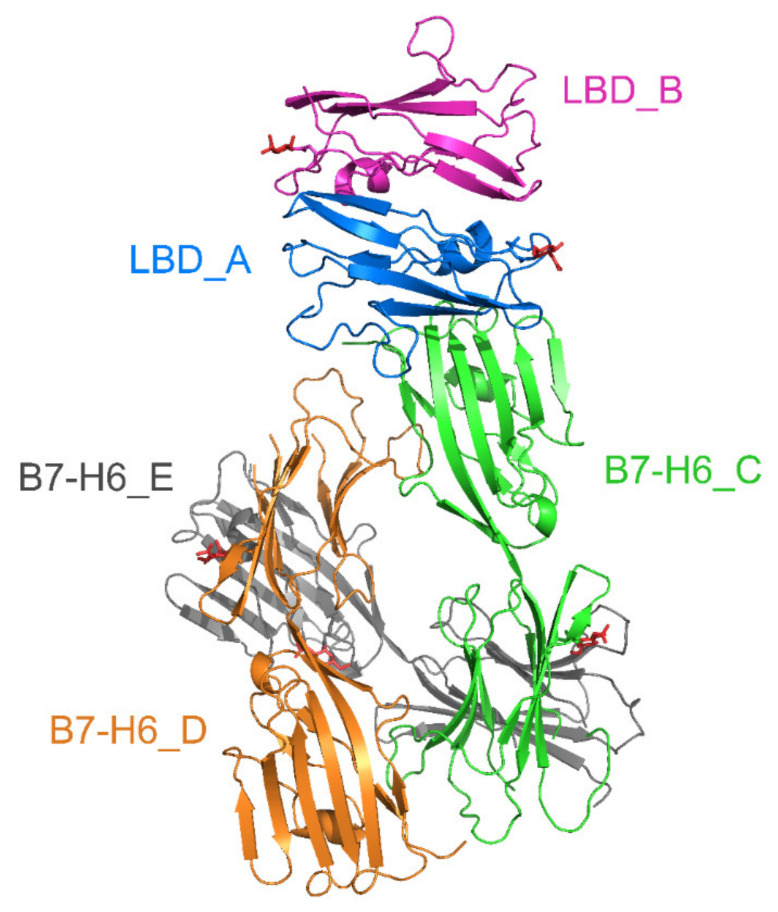
Crystal structure of uniformly glycosylated NKp30_LBD in complex with deglycosylated B7-H6. The asymmetric unit contains two molecules of NKp30_LBD (chains LBD_A and B) and three molecules of B7-H6 (chains B7-H6_C, D and E). Interaction interface between chains LBD_A and B7-H6_C corresponds to the interaction surface observed in the previously published structure of this complex (PDB 3PV6) [20].

**Figure 9 cancers-12-01998-f009:**
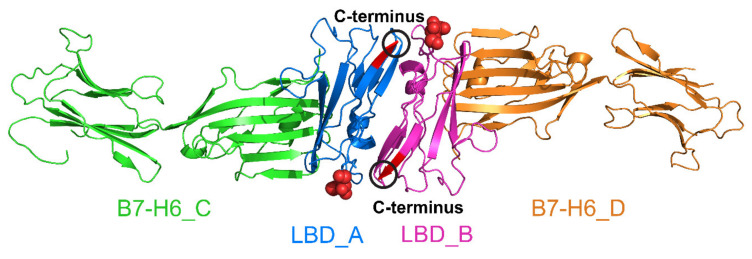
A dimer of the NKp30_LBD:B7-H6 complex is observed in the presented crystal structure. Within the NKp30:B7-H6 complex, both pairs of molecules bind to each other almost identically, and the binding mode of the two proteins is also very similar to that described based on the previously published structure of the complex (PDB 3PV6) [20]. However, the structure of the NKp30_LBD dimer is distinct to the previously published structure of NKp30_LBD itself (PDB 3NOI) [19]. The dimer has a two-fold rotation symmetry, and the *N*-glycosylation site at Asn42 of LBD_A (GlcNAc residue highlighted as spheres) is close to C-terminus of LBD_B (highlighted in red) and vice-versa.

**Figure 10 cancers-12-01998-f010:**
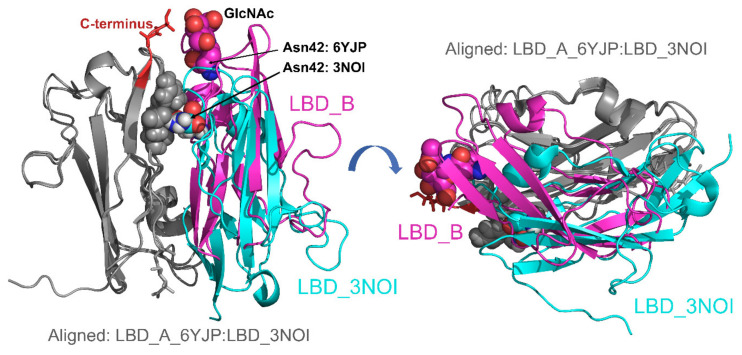
Glycosylation-induced NKp30 dimerization positions the glycans at Asn42 residues near the C-terminal stalk regions. The NKp30_LBD dimers observed in the present crystal structure PDB 6YJP and the PDB 3NOI [19] were aligned using molecules on one side of the dimer only (grey color). On the left —the *N*-glycosylation site at Asn42 of LBD_B, highlighted as spheres, is near the C-terminus of LBD_A (highlighted in red), whereas the Asn42 residue in LBD_3NOI is buried within the dimer interface, interacting with Glu26 and Arg28 (all highlighted as spheres). On the right—side view of the dimer interface showing the difference in arrangement of the two dimers.

**Figure 11 cancers-12-01998-f011:**
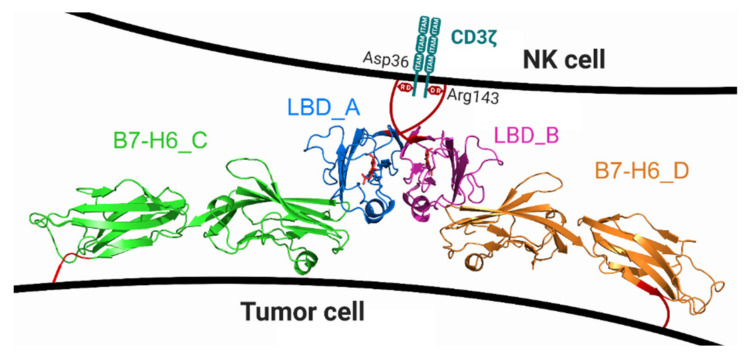
Model of the possible position of the NKp30_LBD:B7-H6 dimeric complex, as observed in the crystal structure (PDB 6YJP), within the NK cell immune synapse. B7-H6 has a very short stalk sequence of several amino acids only, whereas NKp30 has the 15 amino acids long stalk region at its C-terminus (red lines). The stalk is long enough to confer flexibility to the NKp30 ligand binding domain. Such an arrangement would bring the membranes of both cells into very close contact, and such an effect could be further potentiated by NKp30 oligomerization. Local deformation of the NK cell plasma membrane caused by the conformational change of the stalk region induced by ligand binding might trigger signal transduction through the CD3ζ chains associated with the NKp30 transmembrane domain thanks to the interaction of CD3ζ Asp36 residue with NKp30 Arg143 residue occurring at plasma membrane which is required for NKp30 signaling [7].

**Table 1 cancers-12-01998-t001:** Data collection statistics and structure refinement parameters for the NKp30_LBD:B7-H6 crystal structure. Values in parentheses refer to the highest resolution shell.

PDB Code	6YJP
Data processing statistics	
Space group	*C*2
Unit-cell parameters *a*, *b*, *c* (Å); α, β, γ (°)	166.0, 86.5, 111.3; 90, 97.6 90
Resolution range (Å)	48.95-3.1 (3.29-3.1)
No. of observations	99061 (12123)
No. of unique reflections	27102 (3968)
Data completeness (%)	95 (87)
Average redundancy	3.7 (3.1)
Mosaicity (°)	0.09
Average *I*/*σ*(*I*)	5.7 (0.8)
Solvent content (%)	65
Matthews coefficient (Å^3^/Da)	3.54
Wilson B-factor (Å^2^)	105.5
*R* _merge_	0.104 (0.971)
*R* _pim_	0.081 (0.804)
*CC1*/*2*	0.995 (0.645)
Structure refinement parameters	
*R* _work_	0.272
*R* _free_	0.322
*R* _all_	0.275
Average *B*-factor (Å^2^)	155.6
RMSD bond lengths from ideal (Å)	0.005
RMSD bond angles from ideal (°)	1.59
Number of non-hydrogen atoms	6907
Number of water molecules	0
Ramachandran statistics: residues in allowed/favored region (%)	99.5/92.9

$R_{merge}=\sum_{h}\sum_{i}\left|I_{hi}-\langle I_{h}\rangle \right|/\sum_{h}\sum_{i}I_{hi}$, $R_{pim}=\sum_{h}\sum_{i}{\left(n_{h}-1\right)}^{-1/2}\left|I_{hi}-\langle I_{h}\rangle \right|/\sum_{h}\sum_{i}I_{hi}$, and $R=\sum_{h}\left|\left|F_{h,obs}\right|-\left|F_{h,calc}\right|\right|/\sum_{h}\left|F_{h,obs}\right|$, where $I_{hi}$ is the observed intensity, $\langle I_{h}\rangle$ is the mean intensity of multiple observations of symmetry-related reflections, and $F_{h,obs}$ and $F_{h,calc}$ are the observed and calculated structure factor amplitudes. $R_{work}$ is the R factor calculated on 95% of reflections excluding a random subset of 5% of reflections marked as “free”. The final structure refinement was performed on all observed structure factors. RMSD, root-mean-square deviation, PDB, published structure.

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
