# Peer review of "Natural Killer Cell Activation Receptor NKp30 Oligomerization Depends on Its N-Glycosylation"

_cancers, 2020, doi:10.3390/cancers12071998_

Round 1

Reviewer 1 Report

This is a very thorough biochemical and structural analysis that provides new insights into the N-glycosylation-dependent oligomerisation of NKp30 and thus how NK cells may recognise tumour cells expressing B7-H6 ligand and become activated. It is a long paper to read with complex biochemistry and the mechanisms the authors are proposing are not always clear to me. The main finding is that NKp30 oligomerises upon ligand binding (and this is dependent on glycosylation) to induce signal transduction. As such, I think it would really enhance understanding the readers understanding if such a model could be added as the final figure in the main manuscript (pls see comments below)

Comments:

  1. I think Figure S3 should be converted to the final figure in the manuscript. It's really colourful and highlights how the exciting structural findings (dimerisation) may relate to NKp30 signaling. Another major finding of the paper is the stalk/N-glycosylation-dependent oligomerisation and how this may contribute to receptor clustering and signal transduction (Discussion, page 14, line 532) and so I feel the authors should enhance this figure by showing how this initial B7-H6-induced dimerisation followed by subsequent N-glycosylation/stalk region-dependent oligomerisation would potentiate NKp30/B7-H6 signaling because it is an important mechanism for other NK cell receptors and NCR family members.
  2. Are there insights into ligand binding that might be discussed in the context of other NCR receptor family members that also form dimers?

Reviewer 2 Report

In this manuscript, Skorepa et al. studied glycosylated and deglycosylated forms of the activating NK receptor NKp30. Using SEC and AUC, they found that glycosylated NKp30 forms oligomers in solution, and that deglycosylation with EndoF1 reduces these oligomers to monomers. There findings are similar to those reported previously by another group (ref. 21). The difficulty with both studies is that no information is provided regarding the oligomeric state of NKp30 receptors on the cell surface, which is their natural environment. Most likely, glycosylated NKp30 simply aggregates in solution, a not uncommon phenomenon for recombinant proteins. I do not believe these aggregates/oligomers are biologically relevant. 

Reviewer 3 Report

Based on the results from previous publications, the authors studied whether NKp30 oligomerization depends on its N-glycosylation and solved the crystal structure of glycosylated NKp30 with its ligand B7-H6.  They found that NKp30 oligomerization depends on its N-glycosylation. They proposed that glycosylation-mediated dimerization might be necessary for NKp30 stable signal transduction upon B7-H6 binding. Generally, the results are solid and supporting the conclusions.

The following experiment/discussion can be expected:

  1. The authors should show the data wherever cited as “data not shown”.
  2. Mark individual lanes in Figure 2b.
  3. How does purified B7-H6 C212S appears in reducing and non-reducing gels? Simply, the authors can load B7-H6 C212S protein on reducing and non-reducing gels.
  4. It is confusing that NKp30_LBD appears to be monomer in Figure 5a, whereas the authors observed NKp30_LBD dimer in the crystal structure. Moreover, the authors proposed that this dimerization (potentially mediated by glycosylation) was important for the receptor function. More explanation is needed.

Reviewer 4 Report

In the article ‘Natural killer cell activating receptor NKp30 oligomerization depends on its N-glycosylation’ by Skořepa
et al, the authors try to present the N-glycosylation involved NKp30 oligomerization by using overexpression system in both HEK293 and HEK293S GnTI-neg cells as well as the in vitro endoglycosidase F1 digestion system.  How the glycosylation impact activating receptors is important for NK function, however, there are still some issues need to be addressed in this study. 

First, the authors should show data to confirm the difference of N-linked structure among WT, GnTI-, and DEG group (Fig 3).  This is critical evidence to support the N-glycosylation affect NKp30 oligomerization. 

Second, it would be better if the authors provide the impact of mutation on ligand binding site (Asn42 and Asn68) and Ig V-type (Asn43 and Asn 57) on oligomerization by using site direct mutagenesis to reconfirm the importance of N-linked modification. 

Round 2

Reviewer 3 Report

The authors have addressed all my questions.

Reviewer 4 Report

In the article “Natural killer cell activating receptor NKp30 oligomerization depends on its N-glycosylation”, authors demonstrated that N-linked modification highly impact the oligomerization, which is critical for NK effector function.  Importantly, they solved the crystal structure of B7-H7/NKp30 complex, which provide information how the glycosylation affects NK function through ligand-receptor binding affinity.